# Longest-Range UHF RFID Sensor Tag Antenna for IoT Applied for Metal and Non-Metal Objects [note 1]

**DOI:** 10.3390/s19245460

**Published:** 2019-12-11

**Authors:** Franck Kimetya Byondi, Youchung Chung

**Affiliations:** Information and Communication Engineering Department., Daegu University, Kyungsan City 38453, Korea; kimetyafrank@gmail.com

**Keywords:** RFID tag antenna, long range RFID tag, cavity antenna, RFID metal tag, RFID sensors, Internet of things, wireless sensor network, RFID-based IoT, RFID based WSN, smart RFID

## Abstract

This paper presents a passive cavity type Ultra High Frequency (UHF) Radio Frequency Identification (RFID) tag antenna having the longest read-range, and compares it with existing long-range UHF RFID tag antenna. The study also demonstrates mathematically and experimentally that our proposed longest-range UHF RFID cavity type tag antenna has a longer read-range than existing passive tag antennas. Our tag antenna was designed with 140 × 60 × 10 mm^3^ size, and reached 26 m measured read-range and 36.3 m mathematically calculated read-range. This UHF tag antenna can be applied to metal and non-metal objects. By adding a further sensing capability, it can have a great benefit for the Internet of Things (IoT) and wireless sensor networks (WSN).

## 1. Introduction

RFID uses electromagnetic fields to identify automatically and to track tags attached on any object. It’s used for applications like animal tagging, asset tracking, electronic passports, smartcards, store security, logistic and etc. [1]. The tags store object information electronically. There are passive and active RFID tags. There are LF (low frequency), HF (high frequency) and UHF RFID tag antennas [2] based on the frequency bands. Passive tags have no battery and collect energy from nearby RFID readers by interrogating radio waves, whereas active tags have a battery and can be detected over 100 m. 

According to applications and design technology, there are many articles published about UHF active tags. The read-range of the active tag is above 1 km with tag sensitivity under −80 dBm in 5.8 GHz and 433MHz bands [3,4,5]. On the other hand, a general passive UHF tags have about 10–20 m read-range based on the sensitivity of the tag chip and the types of antennas.

Our passive UHF RFID tags have longer read-range than other designs. There will be benefits of tag with longer read-range since it can be incorporate with the sensing and communicating capabilities.

Article [6] describes many novel applications of RFID sensors, novel antennas for metallic surface, 3D antennas, multi-band antennas, omnidirectional, and directional antennas in UHF, HF, or microwaves (MW) frequency bands. The sizes and read-range are also compared with different RFID Chips, and the longest read-range in the article is 14.6 m for the metal mountable antenna. Our passive UHF RFID tag had 26 m read-range using Higgs 4 RFID Chip manufactured by Alien Tech. Higgs 4 has sensitivity of −20.5 dBm.

The more sensitivity of RFID tag chip has longer read-range with the same antenna. Different manufactures and different RFID chips have different level of sensitivities. The chip with the better sensitivity allows the tag to be read from farther range. If tag with longer read-range is integrated with sensors, it can have the following advantages: wireless powered identification in non-line-of-sight way, wide coverage and mobility RFID readers, and the wide range of measurement of sensor tags, such as health monitoring sensor tag [7].

RFID sensing techniques have attracted significant interest. Recent industrial applications include the sensing of temperature and humidity [8], deformation [9], pressure [10], steel cracks and corrosion [11], structure of concrete [12], and monitoring pipeline integrity [13], etc. Other popular applications in healthcare devices include wearable and implanted sensing devices for glucose monitoring [14], blood pressure [15], intraocular pressure [16], on-skin monitoring discrimination of breath anomalies [17], as well as color measurement and analysis [18], etc.

According to the fundamentals of RFID, radio frequency energy harvesting can be divided into inductive coupling and backscattering [19]. The HF RFID working at the carrier frequency 13.56 MHz transmits and receives power with near-field inductive coupling [20], and the UHF working at the carrier frequency 840–960 MHz deals with power transmission and reception with far-field backscattering [21].

In this paper, we are going to focus on a long-range UHF passive tag and compare it to our cavity type tag antenna design. A mathematical proof of read-range is described. Our proposed long-range for metal and non-metal mounted with its long reading range characteristic, 3D geometry can be applied for auto-part logistic, vehicle road distance measurement. Further, if we add humidity and temperature sensing, it can be useful for home automation, health care, food logistic, and temperature as well as humidity monitoring, etc.

The structure of this article is organized as follows. Section 2 presents the necessity of cavity structure design. Section 3 reviews long-range UHF RFID tag antenna and compare it to our proposed design. Section 4 presents our proposed antenna design and its read-range calculation. Finally, Section 5 concludes our work.

## 2. Cavity Structure

The cavity structure is described well in our previously published paper [22], which says that the general label tags attached on metal material cannot operate well, because the impedance of tag antennae can be changed. Previous cavity tags applied to metal environments have used Styrofoam instead of air, the four sides of Styrofoam and the back covered by copper metal to make the cavity. The tag antenna is printed on RF board, the RF board is put on top of Styrofoam, and then a plastic cover is used for protection. In our new cavity tag antenna described in chapter four, the plastic cover is used both for protection and to form the cavity structure. It uses air instead of Styrofoam, the tag antenna is attached on the top-inner side of the plastic cover, four sides of the plastic cover have copper metal attached to it, the back side of plastic cover is empty, and the metal object where the cover will be placed will play the role of back metal to realize the cavity structure. This is just in case our tag is attached on metal environment. Our structure conserves material and increases accuracy in fabrication. In case we attach the proposed tag antenna to a non-metal object, we can put a cooper metal on the back side of the tag antenna to make a cavity, and then attach it to any non-metal object. This will still maintain the advantage of a long read-range since the fabrication became easier and accurate.

In short, our proposed new empty cavity structure has the novelty of reducing material by using air instead of Styrofoam, excluding the use of RF board, using a plastic cover for both protection and to form the cavity, and using the metal object as back metal. Since the plastic cover is made by a 3D printer, this improves the accuracy in fabricating the antenna, and contributes to getting the long reading-range calculated and measured in section four of this paper.

## 3. Review on Long-Range UHF RFID Tag Antenna

In this paper, we focus on UHF RFID long-range passive tag antenna published from 2013 to now. However, even the read-ranges of passive tags published before 2013 are shorter than the proposed one. For a better understanding of existing evidence, we have review the literatures in [23,24,25,26,27,28,29,30,31], and in Table 1.

All the above papers are designed to have long read-range, as we can see in the following table. Each antenna is designed for a specific frequency, and uses a different type of chip. The design is also different. Considering the read-range that every paper shows, it’s clear that our proposed cavity type UHF metal tag has a longer read-range than the previous one, 26 m outdoor and 36.3 m mathematically calculated distance.

Our proposed RFID sensor tag antenna reached an outdoor read-range of 26 m due to the accurate cavity structure fabrication, tuning of parameter ant_h value from 5 to 10 mm using an Alien Higgs 4 chip with a high sensitivity of −20.5 dBm.

## 4. Antenna Design and Read-Range Calculation

In special RFID applications such as attaching the RFID tag antenna to metal components, the general label tag antennae cannot operate well, because the impedance of tag antennae can be changed. Our proposed cavity structure tag antenna resolves this problem and can be readable even when attached to metal objects like a pallet, cart, or car, and other non-metal objects like wood, concrete walls, etc. Figure 1 shows in detail the geometry of the Empty Cavity Tag (ECT) antenna and Table 2 shows the parameter value. The UHF RFID ECT antenna was designed using computer simulation technology (CST) [32]. The relative dielectric constant of air ε = 1, and for PLA plastic cover ε = 1.3 were inserted into CST for design. These pictures are from CST simulation. The first Figure 1a shows the inner shape of the tag antenna and antenna parameters. The parameters ant_h, ant_w are the main part of the antenna, and port_w is used for connection of the tag to the antenna.

The parameter gap is the place where the tag chip will be placed. The remaining parameters make a square with the antenna play the role of T-matching for the antenna and tag. Figure 1b shows the top side of the plastic cover and its dimensions. The yellow metal behind it represents the back copper metal. This metal can represent the object where the tag antenna will be placed. In case we attach the tag antenna to the non-metal object, we can put that back metal to make the cavity structure. Figure 1c shows the inner height of 10 mm of the cavity where the tag chip is placed at the gap. The side copper on the four sides of the cavity will be connected with the back copper via the metal contact (M.C.). The plastic cover (PC) was made with a 3D printer using differently PLA (Polylactic Acid) and ABS (Acrylonitrile Butadiene Styrene copolymer) filaments. Further, as you can see in the inner shape near the gap for connection of the tag, there is a space where we can incorporate the extra sensing unit like humidity or temperature which can be feed with the RF power provide from the two ports where the tag chip is connected. This will help getting sensing unit without traditional battery. Finally, Figure 1d shows the back copper attached at the back of PC to form a cavity structure. Figure 1a–c represents the position of back copper length we used for parameter sweeping of the back metal (BM). This helped us to know the effect of BM when placed behind the plastic cover (PC). When there is no BM, we did not get the proper reflection coefficient S11. When the size of BM as point A in Figure 1d covers the 140 × 60 square space of PC, we got a proper S11. Further, when the size of BM increased as B (170 × 60 mm) and C (200 × 60 mm) in Figure 1d, we still got the proper value of s11 with just a little difference. Since B and C size of BM give proper value of S11, the BM should be B or C size.

Table 2 shows the parameters we used to design and fabricate our proposed longest-range RFID for metal and non-metal tag antenna. All dimensions shown are in millimeter. Parameter tmat means T-matching, and mat means matching, as said above it contributes to the matching between impedance of tag antenna and the impedance of Alien Higgs 4 chip. We achieved a proper S11 value by doing the parameter sweeping of the parameter ant_h. Two values of ant_h respectively 5 and 10 mm gave a promising S11. We used those values and fabricated two different sets of tag antennae. We measured the read-range outdoors and with an anechoic chamber.

### 4.1. Impedance Matching and Antenna Gain

Figure 2 shows the parameter sweeping simulation result of reflection coefficient S11 based on the change of ant_h values. The optimized value with ant_h = 5 at 920 MHz S11 was −12.138 dB. We fabricated the antenna and measured the reading range which was just 8 m. And we fabricated the antenna with ant_h = 8, 10 resonating at 0.97 GHz, the read range was super high 26 m at 0.92 GHz. We wanted to know what can be the reason of this situation. We then prove mathematically the value of read range of ant_h = 5, and ant_h = 10 in the following section.

A proper antenna conjugate matching to the RFID chip was found. The return coefficient S11 was calculated using Equation (1). Alien Gen2 Higgs 4 RFID chip was used.
(1)$$S_{11}=20\;\log_{10}\left(\frac{Z_{a}-Z_{c}^{*}}{Z_{a}+Z_{c}}\right),$$
where, $Z_{a}$ is antenna impedance, $Z_{c}$ is chip impedance and $Z_{c}^{*}$ is the conjugate chip impedance.
(2)$$Z_{c}=\frac{R}{1+\omega^{2}R^{2}C^{2}}-j\frac{{\omega R}^{2}C}{1+\omega^{2}R^{2}C^{2}},$$
where, $Z_{c}$ is the chip impedance, ω the frequency, R the chip resistance, C is the chip capacitance. These values are provided in the Alien Higgs 4 data sheet. Figure 3 shows the schema of the Higgs 4 chip and different value. The R value is 1800 ohm, C value is 0.95 pF, and resonance frequency is ω=2πf. F is 920 MHz for ant_h = 5.

In Equation (1), the simulation antenna impedance value is calculated using this complex $Z_{a}=a+jb$, a is the real part and b is the imaginary value. At 920MHz, $Z_{a}=12.076+j175.89$. $Z_{c}$ is the chip impedance. Using Equation (2), the chip impedance is $Z_{c}=18.258-j180.34$ at 920 MHz with ant_h = 5. And the conjugate chip impedance is $Z_{c}^{*}=18.258+j180.34$. Using all these values the value of Equation (1) becomes $S_{11}=-12.094\;\mathrm{dB}$. Same operation was done for the ant_h = 10. This gave us its proper value omitted here.

### 4.2. Read-Range Calculation

Read-range is the most important tag performance characteristic. Figure 4 shows the simplified RFID system with Rmax as the distance between the reader and tag antenna. Page 349 and 350 of the [2] and papers [33,34,35,36] show that the read-range can be calculated using Friis’ uplink model Equation (3) considering that the reader and tag polarization are well aligned.
(3)$$Rmax=\frac{\lambda }{4\pi }\sqrt{\frac{P_{t}G_{t}G_{r}\tau }{P_{th}}},$$
where λ is the wavelength calculated using the speed of light $c=3\times 10^{8}$ m/s divided by frequency f = 920 MHz. For ant_h = 5.
(4)$$\lambda =\frac{3\times 10^{8}}{920\times 10^{6}}=0.326\;m,$$
where $P_{t}$ And $G_{t}$ are respectively reader’s power transmission and gain. The Effective Isotropique Radiated Power (EIRP) is given by the product of $P_{t}$ and $G_{t}$. We used an ALR−9900 RFID reader which has RF power max 4 watts EIRP with Alien antenna. So, Equation (3) becomes:
(5)$$r=\frac{\lambda }{4\pi }\sqrt{\frac{EIRP\times G_{r}\times \tau }{P_{th}}},$$
where $G_{r}$ Is the tag antenna gain. The gain is 6.43 dBi as stated in the simulation result of gain pattern in Figure 3 above. The gain can be written as:
(6)$$G_{r}=10^{\frac{6.43}{10}}=4.395\;mw$$
$P_{th}$ is the chip’s sensitivity, in other words, the minimum threshold power for a tag to respond to the reader’s request. We used alien’s Higgs 4 chip which operates in the frequency range of 840–960 MHz and have −20.5 dBm sensitivity during read. So $P_{th}$ becomes:
(7)$$P_{th}=10^{\frac{-20.5}{10}}=8.913\;uW,$$τ is the power transmission coefficient and determines how good the tag antenna and chip are matched. It is given by the Equation (8):
(8)$$\tau =\frac{4R_{c}R_{a}}{{\left[Z_{c}+Z_{a}\right]}^{2}}=0.938,\;0\leq \tau \leq 0$$
where, $R_{c}$ is the chip resistance and
$R_{a}$ is the antenna resitance, the chip impedance is $Z_{c}=R_{c}+jX_{c}$ and the antenna impedance is $Z_{a}=R_{a}+jX_{a}$. At a given frequency if the factor τ=1, it means that the chip and antenna are perfectly matched. ${\left[Z_{c}+Z_{a}\right]}^{2}$ Can be calculated using the Equation (9):
(9)$${\left[Z_{c}+Z_{a}\right]}^{2}={\left(R_{a}+R_{c}\right)}^{2}+{\left(X_{a}+X_{c}\right)}^{2},$$

Note that, $Z_{a}$, $R_{a}$, $X_{a}$ are CST simulation data. And the calculation here uses antenna simulation data for ant_h = 5. The calculation of simulation value of ant_h = 10 was omitted, but only final results are presented in this paper, and the calculation logic is same. $Z_{c}$, $R_{c}$, and $X_{c}$ are based on the chip data sheet and calculated using Equation (2).

Using Equation (5) the mathematical simulation read-range (SRR) for ant_h = 5 was 32.77 m at a frequency of 920 MHz, while the outdoor measured read range was 8 m with Alien RFID reader. At 970 MHz, the simulation mathematical result of ant_h = 10 was 36.3 m, while the measured read-range (MRR) was ~25 m. The outdoor measurement with the Alien RFID reader at the 920 MHz was 26 m. The difference might result in a mismatch in the form of fabrication and polarization misalignment. The result with anechoic chamber also gave 26 m for ant_h = 10.

For a greater understanding, Figure 5 shows the read-range according to the variation of frequency. The black graph shows the simulation calculation of ant_h = 5 at 920 MHz. The green graph shows the measured calculated using measured $Z_{a}$, $R_{a}$, and $X_{a}$ for ant_h = 10. As you can see, with simulation results ant_h = 5 has RR = 32.77 m, while measured one (omitted) was 8 m. Further, ant_h = 10 which in simulation resonated at 970 MHz, gives a measured RR of ~25 m at 920 MHz. These $Z_{a}$, $R_{a}$, $X_{a}$ were measured with Agilent’s E5071B network analyzer in our lab. The RR for ant_h = 10 outdoor measurement at 920 MHz with Alien reader antenna was 26 m, and the same read-range from the measurement in an anechoic chamber. So, we consider our ant_h = 10 with read-range 26 m as a longest read range RFID tag antenna.

### 4.3. Reader-To-Tag and Tag-To-Reader RF Power Transfer

The passive RFID sensor tag receives the reader’s power to complete the sensing and data transmission process. The power density S of RF energy at a read distance r from the reader’s antenna can be calculated using the mathematical Friis Equation (10) [37].
(10)$$S=\frac{P_{t}G_{t}}{4\pi r^{2}}=\frac{EIRP}{4\pi r^{2}},$$

The effective size of a tag antenna is $A_{e}$ defined as Equation (11)
(11)$$A_{e}=\frac{\lambda^{2}G_{r}}{4\pi },$$

Then, the received power $P_{\mathrm{tag}}$ can be expressed as Equation (12)
(12)$$P_{\mathrm{tag}}=S\times A_{e}$$

In Equation (13), the RF power reflected by the tag antenna P_tag_ is directly proportional to the tag’s Radar Cross-Section (RCS) σ, where S is the power density. In Equation (14), the power density is S_back_, where P_tag_ and G_tag_ are the transmission power and gain of tag antenna. In Equation (15) G_reader_ is the gain of the reader antenna, the effective dimension of the receiving antenna is Aw = G_reader_
$\lambda^{2}/4\pi$, and the power received by the reader is P_reader_ [38]:(13)$$P_{\mathrm{tag}}=S_{\sigma }=\frac{P_{\mathrm{tag}}P_{\mathrm{tag}}}{4\pi r^{2}}\sigma ,$$
(14)$$S_{\mathrm{back}}=\frac{P_{\mathrm{tag}}P_{\mathrm{tag}}}{{\left(4\pi \right)}^{2}r^{4}}\sigma ,$$
(15)$$P_{\mathrm{reader}}=S_{\mathrm{back}}A_{w}=\frac{P_{\mathrm{tag}}P_{\mathrm{tag}}\lambda^{2}G_{\mathrm{reader}}}{{\left(4\pi \right)}^{3}r^{4}}\sigma ,$$

It’s clear that the gain of the reader and tag antennas, the distance between reader and tag, and the tag antenna’s RCS are the most important parameters to determine the reading distance and efficiency of RFID sensor results. A tag antenna with sensing capability will transmit to the reader the tag’s information, including the tag ID and sensor information.

### 4.4. Fabrication and Result of Long-Range RFID Tag

Figure 6 shows respectively the fabricated ECT RFID tag antenna inner and back cooper views. Antenna, side copper and back copper all are attached on PLA plastic to form the cavity. The PLA plastic was fabricated using 3D printer. We used this fabricated antenna and measured antenna resistance $Z_{a}$, $R_{a}$, $X_{a},$ and gain with network analyzer, then we used the data to calculate the read-range.

The Figure 7 shows the outdoor read-range pattern of the fabricated ant_h = 10 antenna. We measured the read-range by turning the tag antenna angle according to Phi and Theta directions. When the antenna is turned 180 degree so that its back side faces the reader antenna, the reading range is minimum 2.8 m. When the tag antenna is facing the reader antenna at 0 degree, the reading range is maximum about 26 m. Results for other angles 30, 60 etc. are shown in Figure 7. The read-range for 920 MHz with ant_h = 5 has similar read-range pattern as Figure 7, and the measured best reading distance was 8 m. According to Figure 2, the antenna with the size ant_h = 5 should give best results, but the antenna with the size ant_h = 10 gives better results, since the impedance of antenna is changed by resistance value of bonding glue between the copper and the tag chip that attached on the PLA cover.

The read-range pattern has the similar shape with the simulation gain pattern in Figure 8. It means that the tag antenna is detected at long distance only when it is well polarized and faced the reader antenna.

The beam pattern of ECT RFID tag antenna has directivity Figure 8. Due to the metal cavity, at 920 MHz the gain of antenna is 6.45 dBi, the direction of the main lobe 0.0 degree (blue line) and the 3 dB angle in the main direction is 124.6 degree (sky blue line). These results are obtained in CST after simulation.

Figure 9 shows the graph of simulated and fabricated ant_h = 10 antenna. Simulation reflected coefficient S11 is −12.6 dB. After fabrication of the antenna, we measured the reflected coefficient using Agilent’s E5071B network analyzer. The antenna presents a measured S11 of −17.4 dB at 970MHz because there is no RFID chip and bonding resistance between tag antenna and tag chip. We emphasize that ant_h = 10 resonates at 970 MHz for simulation and measurement because the final tag with the RFID chip will resonate at 920MHz including the extra resistance values for fabrication. Therefore, the measured the read-range at 920 MHz with Alien Reader antenna is 26m which is the best.

## 5. Conclusions

This work presents a comparative study of existing long-range UHF RFID tags. Further, we demonstrate mathematically and experimentally that our proposed (ant_h = 10) long-range UHF RFID cavity type tag antenna has a longer read-range than existing passive tags. Our tag was designed to be attached on metal and non-metal objects with 140 × 60 × 10 mm^3^ size and reached 26 m measured read-range and 36.3 m mathematical simulated, calculated read-range, which is longer than existing passive tag antennae. With this result, we have improved the previous paper [22] and proven why the read-range is longer. The difference in results is based on the fabrication accuracy and misalignment problem. The back copper of our antenna can be replaced by the metal on which the tag antenna will be attached. In case we attach the tag antenna to a non-metal object we need to put the back copper too. So, it saves material in case of metal object. It has also the advantage of a long-range for non-metal objects.

We will provide many benefits by adding a sensing unit like humidity and temperature to our longest read-range tag antenna. Likewise a sensor with no battery can be used in sensor networks and IoT systems. This sensor tag antenna will play the role of feeding power to the sensing unit. Network size can be reduced in terms of the reader’s quantity grace using this advantage of the long-range RFID tag antenna. Such opportunities will be pursued in further work.

## Figures and Tables

**Figure 1 sensors-19-05460-f001:**
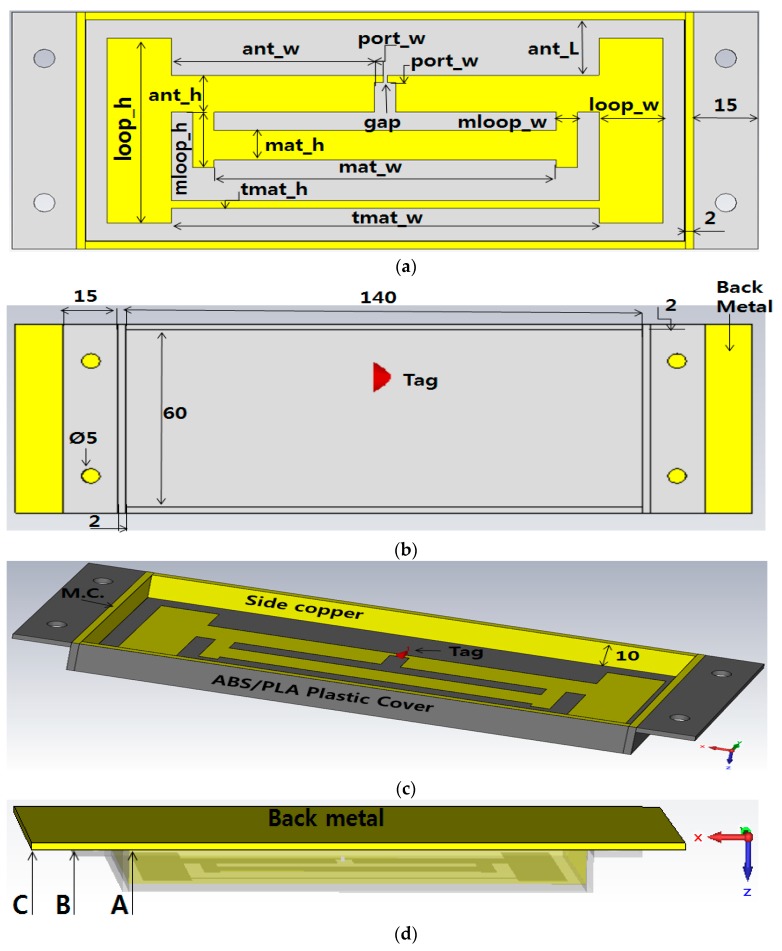
Cavity tag antenna 3D geometry, (**a**) bottom view; (**b**) top view; (**c**) side view; (**d**) back view.

**Figure 2 sensors-19-05460-f002:**
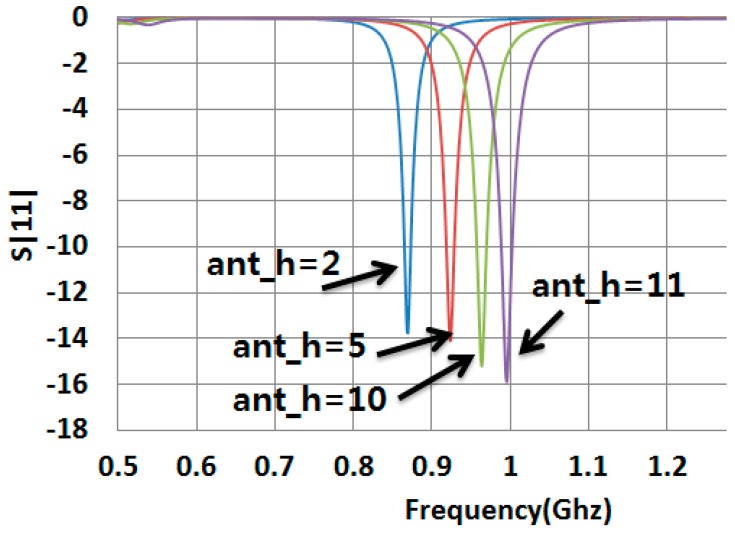
S11 vs. Parameter ant_h sweeping of the tag antenna.

**Figure 3 sensors-19-05460-f003:**
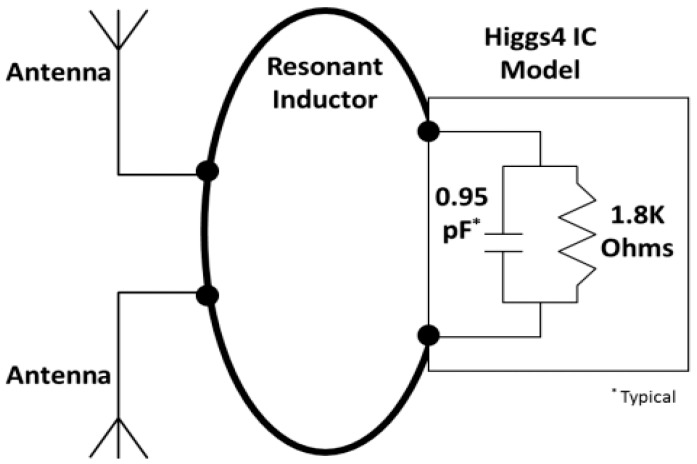
Higgs 4 chip application model (from Alien Higgs 4 data sheet).

**Figure 4 sensors-19-05460-f004:**
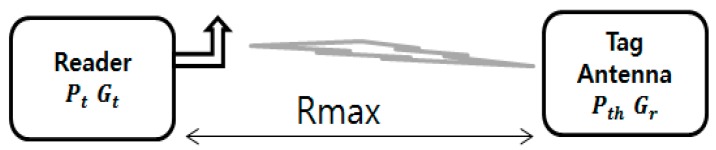
Diagram of simplified RFID system.

**Figure 5 sensors-19-05460-f005:**
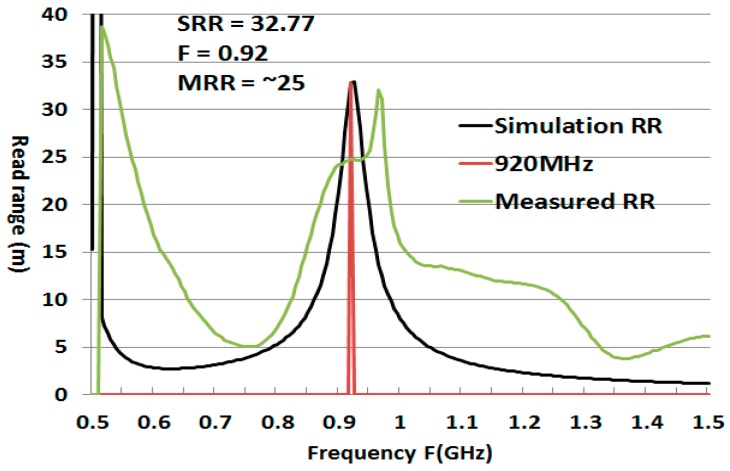
Calculated and measured read-range (RR) variation with frequency.

**Figure 6 sensors-19-05460-f006:**
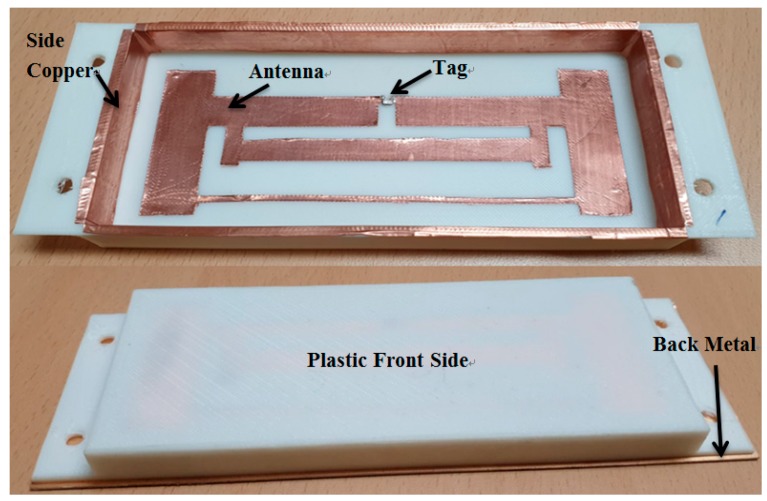
Fabricated empty cavity tag antenna.

**Figure 7 sensors-19-05460-f007:**
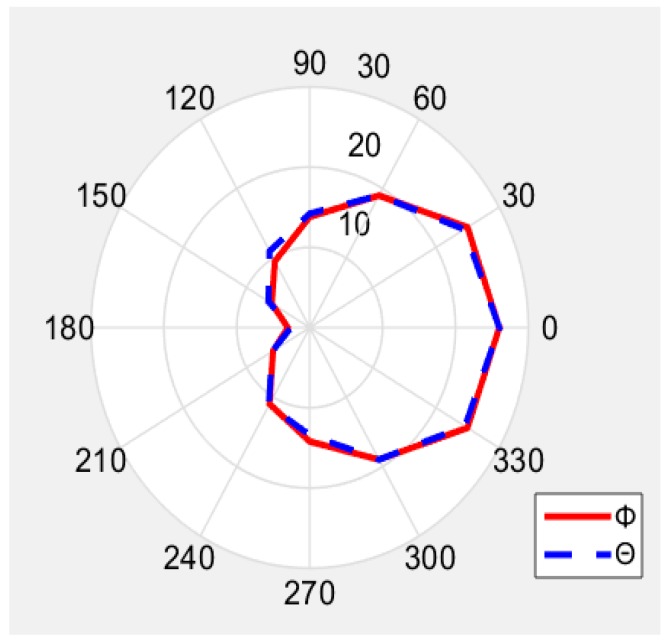
Measured read-range pattern in (m).

**Figure 8 sensors-19-05460-f008:**
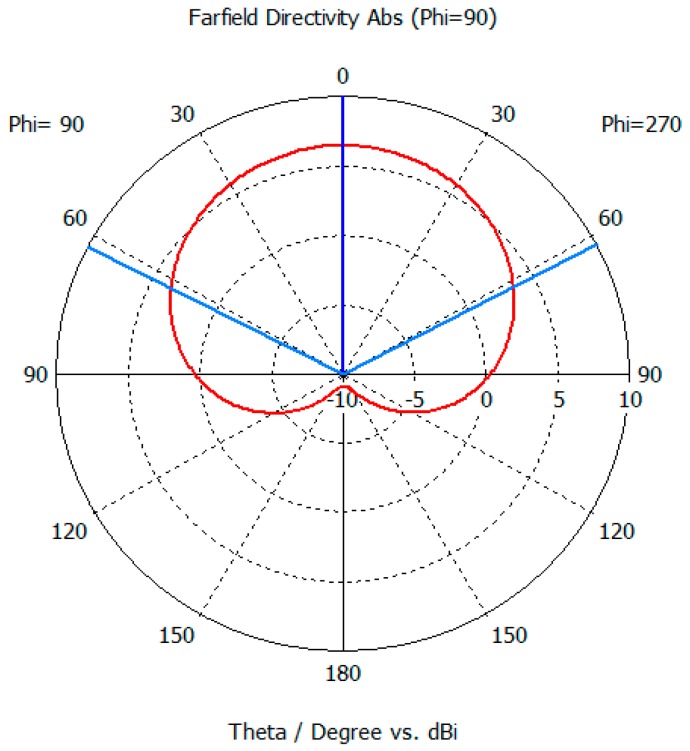
Simulated antenna pattern.

**Figure 9 sensors-19-05460-f009:**
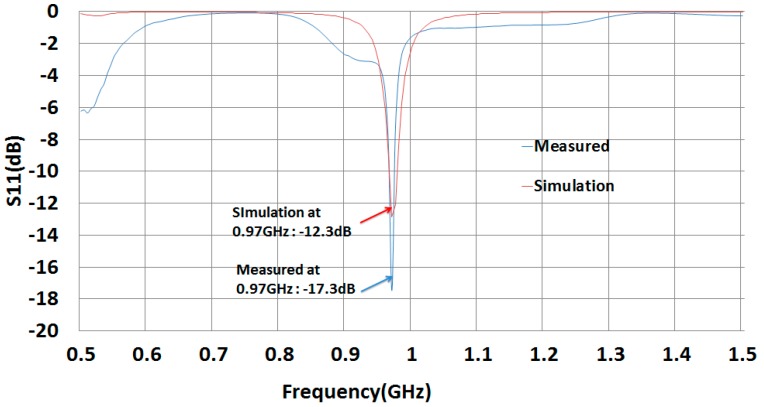
S11 of simulated and fabricated tag antenna.

**Table 1 sensors-19-05460-t001:** Long range RFID antenna comparison from 2013–2019.

No	Reference	Frequency (GHz)/Size (mm)	Read-Range (m)	Tag or Chip/Sensitivity	Technology
1	Paper [23]	0.908/155 × 110	10	Not described	Metal tag antenna
2	Paper [24]	0.920–0.925/credit card size; 0.860–0.960/credit card size	10; 5	Alien-9768	Alien-9768
3	Paper [25]	9.75/NA	10	RCS of 21 dB	Modulated scattering
4	Paper [26]	6.2/NA	Simulated	Chip less	Chip less
5	Paper [27]	0.868/Na	15	EM4324	L-matched RFID tag
6	Paper [28]	0.878/Na	Simulated	Not described	Planar metal tag
7	Paper [29]	0.866/89.5 × 25	18	Higgs 4 −20.5 dBm	Wide band tag antenna
8	Paper [30]	0.9125/41 in diameter and 6.48 thick	16.7	−18 dBm	C-shaped loop
9	Paper [31]	0.915, 0.92/140 × 60 × 10	12	Higgs 3/−20 dBm	Cavity and bottom metal
10	Proposed	0.915, 0.92/140 × 60 × 10	26	Higgs 4 −20.5 dBm	Cavity type Metal tag

**Table 2 sensors-19-05460-t002:** Tag antenna parameters.

Parameters	Size (mm)	Parameters	Size (mm)
ant_w	47.5	cavity height	10
ant_h	5–10	gap	1
tmat_h	2	circle diameter	5
tmat_w	100	mat_h	8
loop_h	50	mat_w	80
loop_w	15	mloop_h	20
port_w	2	mloop_w	5
ant_L	15

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
