# Peer review of "Longest-Range UHF RFID Sensor Tag Antenna for IoT Applied for Metal and Non-Metal Objects"

_sensors, 2019, doi:10.3390/s19245460_

Round 1

Reviewer 1 Report

The authors need to review the paper carefully to ensure that there is no error in it. It is also needed to discuss the results more in detail.

Author Response

Hi dear Reviewer

Please check the rectified uploaded file. and then give please a comment.

Thanks 

Reviewer 2 Report

The long-range UHF RFID antenna presented in this paper is of interest to many IoT applications. Both theoretical analysis and practice are conducted in this paper. The following issues are suggested to improve the quality of this paper: 1. Section 2, lines 87-88: “We used most of Nikitin’s paper technique and formula as described in part four B of this paper, to design and calculate the reading range of our long-range cavity antenna.” The authors need to highlight the novelty of the work presented in this paper. Namely, what are the now design and implementations. 2. In the comparison of the long-range RFID antennas and experimental study of the proposed antenna, the authors need to give the details of the setting how the 26m is obtained. 3. This paper is not written carefully. There are too many grammar problems even in the abstract, such as, (1) Line 9, Ultra High Frequency (UHF) and Radio Frequency Identification (RFID) (2) Also demonstrate…, this is academic writing (3) Line 11, than existing passive tag antenna-> antennas (4) Line 14, Passive tag->passive tags (5) Line 14, metal and non-metal objet->object

Author Response

The long-range UHF RFID antenna presented in this paper is of interest to many IoT applications. Both theoretical analysis and practice are conducted in this paper. The following issues are suggested to improve the quality of this paper:

Section 2, lines 87-88: “We used most of Nikitin’s paper technique and formula as described in part four B of this paper, to design and calculate the reading range of our long-range cavity antenna.”

The authors need to highlight the novelty of the work presented in this paper. Namely, what are the now design and implementations.

Answer : Line 84-88 è In short our proposed new empty cavity structure has novelty of reducing material by using air instead of Styrofoam, excluding the use of RF board, using plastic cover for both protection and to form the cavity, using the metal object as back metal. Since the plastic cover is made by 3D printer, this improves the accuracy in fabricating the antenna, and contributes in getting the long reading-range calculated and measured in section four of this paper.

In the comparison of the long-range RFID antennas and experimental study of the proposed antenna, the authors need to give the details of the setting how the 26m is obtained.

Answer: Line 104-106 è Our proposed RFID sensors tag antenna reached 26m outdoor reading range due to the accurate cavity structure fabrication, tuning of parameter ant_h value from 5mm to 10mm, using Alien Higgs 4 chip with high sensitivity -20.5dbm.

And Line 228-236 è For more understanding, Figure 5 shows the read-range according to the variation of frequency. The black graph shows the simulation calculation of ant_h=5 at 920MHz. The green graph shows the measured calculated using measured, ,   for ant_h=10. As you can see, mathematically with simulation result ant_h=5 has RR = 32.77m, while measured one (omitted) was 8m. And ant_h=10 which in simulation resonated at 970MHz, gives a measured RR of ~25m at 920MHz. These, ,   were measured with Agilent’s E5071B network analyzer in our lab. The RR for ant_h=10 outdoor measurement at 920MHz with Alien Reader antenna was 26m and same result for anechoic chamber(omitted). So we consider our ant_h=10 with read range 26m as a longest read range RFID tag antenna.

This paper is not written carefully. There are too many grammar problems even in the abstract, such as,

(1) Line 9, Ultra High Frequency (UHF) and  Radio Frequency Identification (RFID)

Corrected: line 9

(2) Also demonstrate…, this is academic writing Corrected

(3) Line 11, than existing passive tag antenna-> antennas Corrected

(4) Line 14, Passive tag->passive tags Corrected

(5) Line 14, metal and non-metal objet->object Corrected

Reviewer 3 Report

The subject matter is interesting. However, the quality of the paper should be improved.

The authors described in the introduction a new RFID tag with a better performance that previous prototypes, but a comparison between their prototype and the previous one is missing. Abstract should include the main findings and less details. The English language needs to be improved. Several grammatical errors are observed throughout the manuscript. For example in line 78 the sentence “..the tag antenna is printed is attached on top of Styrofoam” is not well written. In line 108, the authors talked about the dielectric constant instead of relative dielectric constant. In the reference, a manual of the CST is missing. In figure 2, if the legend is included it is not necessary to specify a detail of each S11 results. In figure 5, the axis legend must be turned round. In line 228 after dot is necessary to write a capital letter. The measurement of the antenna impedance indicated in line 232 has been done with or without anechoic chamber. The format of the figures 6,7 and 8 caption is not correct. The results shown in figure 8, how can the authors obtain them? In line 293, the authors detailed that the results presented in the paper are better than the previous works, but a comparison that confirm this assertion is missing.

Author Response

The authors described in the introduction a new RFID tag with a better performance that previous prototypes, but a comparison between their prototype and the previous one is missing.

 Abstract should include the main findings and less details.

Answer: Line 16 : Reduced the abstract contents by removing the folowing parts.

It can be applied for car traffic distance sensing, in case of adding temperature and humidity sensing on it, it can be useful for smart home, healthcare, food logistic and humidity and temperature monitoring system.

The English language needs to be improved.

Several grammatical errors are observed throughout the manuscript. For example in line 78 the sentence “..the tag antenna is printed is attached on top of Styrofoam” is not well written.

Corrected like in line 74 è “the tag antenna is printed on RF board, the RF board is put on top of Styrofoam”

In line 108, the authors talked about the dielectric constant instead of relative dielectric constant.

Answer: Corrected è line 111

In the reference, a manual of the CST is missing.

Answer: added the web site of cst as refrence [31]. Line 111 and line 371

In figure 2, if the legend is included it is not necessary to specify a detail of each S11 results.

Correted è line 173

In figure 5, the axis legend must be turned round. Correted è line 225

In line 228 after dot is necessary to write a capital letter. Correted è line 229

The measurement of the antenna impedance indicated in line 232 has been done with or without anechoic chamber. Answer : line 233 è These, ,   were measured with Agilent’s E5071B network analyzer in our lab.

The format of the figures 6,7 and 8 caption is not correct. Corrected

The results shown in figure 8, how can the authors obtain them? Answer: Line 260-265 è The Figure 7 shows the outdoor read range pattern of the fabricated ant_h =10 antenna. We measured the read range by turning the tag antenna angle according to Phi and Theta directions . When the antenna is turned 180 degree so that its back side faces the reader antenna, the reading range is minimum 2.8m. When the tag antenna is facing the reader antenna at 0 degree, the reading range is maximum about 26m. Results for other angles 30, 60 etc. are shown in the graph.

In line 293, the authors detailed that the results presented in the paper are better than the previous works, but a comparison that confirm this assertion is missing.

Answer: Line 93-103: …. Our proposed RFID sensors tag antenna reached 26m outdoor reading range due to the accurate cavity structure fabrication, tuning of parameter ant_h value from 5mm to 10mm, using Alien Higgs 4 chip with high sensitivity -20.5dbm.

Round 2

Reviewer 3 Report

I confirm that the subject matter is very interesting. The authors have included a lot of changes. I appreciate the great effort of the authors.

However, the quality of the paper has to be improved.  

The authors described in the introduction a new RFID tag with a better performance that previous prototypes which has been published in a Special Issue of Sensors journal. But the only difference are the materials use to make RFID tag. I can’t see the advantages in the research. The English language needs to be improved. Several grammatical errors are still throughout the manuscript. The paper is not well written. In the introduction, the authors wrote (p. number) several times. I don’t understand the reason or objective. A minor detail is that some units are not well written, such as in line 43.

Author Response

The authors described in the introduction a new RFID tag with a better performance that previous prototypes which has been published in a Special Issue of Sensors journal. But the only difference are the materials use to make RFID tag. I can’t see the advantages in the research. 

==> Line 84-88

In short our proposed new empty cavity structure has novelty of reducing material by using air instead of Styrofoam, excluding the use of RF board, using plastic cover for both protection and to form the cavity, using the metal object as back metal. Since the plastic cover is made by 3D printer, this improves the accuracy in fabricating the antenna, and contributes in getting the long reading-range calculated and measured in section four of this paper.

==> Line 100-103

Our proposed RFID sensor tag antenna reached the outdoor read-range of 26m due to the accurate cavity structure fabrication, tuning of parameter ant_h value from 5 to 10mm, using Alien Higgs 4 chip with high sensitivity -20.5dBm.

==> in term of research : line 230-238

We simulated ant_h=10 at 970MHZ, then we  fabricated and measured at 920MHz and prove mathematicaly why we got the long read- range 26m. The research shows that tuning the simulation resonating frequancy at higher frequency, can lead to a great read range result with accurate fabrication.

The English language needs to be improved. Several grammatical errors are still throughout the manuscript. The paper is not well written.

==> Re-read the full manucript and corrected.

In the introduction, the authors wrote (p. number) several times. I don’t understand the reason or objective. A minor detail is that some units are not well written, such as in line 43.

==> Corrected